# Extracting Daily Routines from Raw RSSI Data

**DOI:** 10.3390/s25092745

**Published:** 2025-04-26

**Authors:** Raúl Montoliu, Emilio Sansano-Sansano, Marina Martínez-García, Sergio Lluva-Plaza, Ana Jiménez-Martín, José M. Villadangos-Carrizo, Juan Jesús García-Domínguez

**Affiliations:** 1Institute of New Imaging Technologies, Jaume I University, 12071 Castellón de la Plana, Spain; esansano@uji.es; 2Department of Maths, Jaume I University, 12071 Castellón de la Plana, Spain; martigar@uji.es; 3Department of Electronics, Escuela Politecnica Superior, University of Alcala, 28805 Alcalá de Henares, Spain; sergio.lluva@uah.es (S.L.-P.); ana.jimenez@uah.es (A.J.-M.); jm.villadangos@uah.es (J.M.V.-C.); jjesus.garcia@uah.es (J.J.G.-D.)

**Keywords:** health and wellness applications, monitoring and modeling of human motion, wearable-based systems, location-based services and applications

## Abstract

Detecting behavioral routines is an important research area with many implications in various practical applications. One such application involves studying the behavior of older adults residing in care homes. This paper proposes a comprehensive methodology for extracting and analyzing the daily routines of older adults in care homes. The methodology utilizes raw data comprising signal strength measurements obtained from smartwatches worn by six volunteers over five months. To establish the basis for estimating daily activities, fingerprint-based localization techniques are employed to track the minute-by-minute location of each volunteer. Subsequently, the activity performed by each volunteer is estimated for each day. Finally, the study estimates the probability of a user undertaking each one of the studied activities on a given weekday.

## 1. Introduction

Human activity monitoring holds significant importance for older adults, as it enables caregivers and healthcare professionals to ensure their well-being, safety, and independence. By monitoring the activities of older adults, it becomes possible to detect any potential health issues, changes in behavior, or signs of decline. This monitoring can include tracking physical activity, sleep patterns, medication adherence, and daily routines. By analyzing these data, caregivers can identify any deviations or anomalies that may require attention or intervention. For example, monitoring physical activity can help caregivers detect a decrease in mobility, which could indicate the need for additional support or interventions to prevent falls. Moreover, activity monitoring can provide reassurance to family members, knowing that their loved ones are being looked after and that any changes or emergencies will be promptly addressed.

This paper focuses on obtaining the daily routines of older adult volunteers residing in a care home setting. The methodology presented aims to extract valuable insights from input data comprising RSSI (Received Signal Strength Indicator) measurements obtained via smartwatches worn by the volunteers. These measurements correspond to a network of BLE (Bluetooth Low Energy) beacons deployed throughout the care home. The proposed methodology consists of three key steps. Firstly, the localization of each user is estimated minute by minute throughout the day, utilizing an indoor localization fingerprinting-based method. This step provides essential spatial context for analyzing activity patterns. Secondly, a feature vector is created to describe the activities undertaken by each volunteer at different time intervals within a day. This allows for a comprehensive representation of the daily routines. Lastly, the probability of a specific activity being performed by each volunteer is estimated based on the day of the week. This analysis provides insights into the temporal patterns of various activities.

In summary, the most important contributions of this work are as follows:We derive the daily routines of a set of volunteers from RSSI data by applying fingerprinting-based indoor localization and routine extraction techniques.For each user, we obtain the probability of performing a particular activity on a given weekday and compare this prediction with the user’s actual behavior, as observed by their caregivers.

The data utilized for evaluating the proposed approach are sourced from an openly accessible dataset [1], encompassing RSSI signals from a network of BLE beacons deployed in a nursing home. These signals were recorded by a smartwatch worn by six volunteers throughout their daily activities. The dataset spans nearly five months, and the volunteers involved are aged 84 or older. It is crucial to underscore that the data employed are authentic, with no specific directives given to the volunteers regarding their behavior during the data collection period. They simply proceeded with their daily routines. Furthermore, the dataset incorporates supplementary information provided by caregivers regarding the volunteers’ schedules. Despite potential challenges such as missing data on days when caregivers are on holiday or during their free days, the proposed method sufficiently captures the behavioral patterns exhibited by the volunteers over the study period.

The rest of the paper is organized as follows: Section 2 briefly summarizes some of the most relevant papers in the related research field. Section 3 explains all the methods used in this paper to obtain the daily routine. The dataset used is presented in Section 4. Section 5 details the experiments performed and comments on the results obtained. Finally, Section 6 presents the main conclusions arising from this work and offers insight into potential directions for future research.

## 2. Background and Related Work

Healthcare professionals use the term “activities of daily living” (ADLs) to refer to routine self-care tasks performed at home, outdoors, or in both settings. In elderly care, ADLs are commonly used to assess an individual’s level of functional independence. ADLs are typically represented as a tuple of the following four elements: the activity performed (e.g., cooking, watching TV, sleeping), the start time, the end time, and the location where the activity took place. However, working with all four elements is not always feasible. Identifying the activity performed can be particularly challenging, often requiring manual labeling in ADL datasets. In many studies, including this paper, activities are inferred from the location. For instance, in a dining room, one would expect eating-related activities, while in a gym, physical exercises involving movement are more likely.

Gao, Tan, and Setchi proposed a method for learning ADL daily routines using representations of ADLs as tuples [2]. Their approach employed spatiotemporal neural networks and was evaluated on both a synthetic dataset and the real-world CASAS dataset. The CASAS dataset, published by Washington State University [3], contains long-term sensory data collected from multiple testbeds with real human participants. Widely used in the ambient intelligence field, it is a key resource for ADL-related research. Another notable dataset is ActionSense [4], a multimodal dataset and recording framework focused on wearable sensing in kitchen environments. It provides synchronized data streams and ground truth labels, supporting learning pipelines that analyze human interactions with the physical world during daily kitchen activities.

Previous methods presented in the literature differ in the sensors used to obtain the data and the method applied to estimate the routines. Electricity consumption has been gaining popularity in the last few years. In [5], the authors explored electricity consumption patterns by clustering techniques, as users in the same cluster exhibit similar consumption patterns, represented by their similar daily routines and peak demand periods. Their results demonstrated the substantial heterogeneity in the consumption behavior of the consumers at different times of the day, along with the effect of seasons and non-working days. R. Askari et al. [6] developed a method for detecting meal consumption and physical activity based on blood glucose concentration, measured using continuous glucose monitoring. They used RNN and LSTM neural networks to detect the events that affected blood glucose concentration.

Many papers have used wearable devices, such as mobile phones or smartwatches, as sources of data [7]. Martín et al. [8] presented a system to detect changes in daily routines within a controlled environment, such as a sensorized home, based on an indoor symbolic location system. This system comprised a network of low-cost, easy-to-install Bluetooth Low Energy (BLE) transmitter beacons and a mobile receiver. The user’s symbolic location was estimated using the Received Signal Strength Indicator (RSSI) and the K-Nearest Neighbour (KNN) model, which is merged with the acceleration provided by the receiving mobile device. The location is used to estimate the time spent in each monitored room to infer a time-based routine. Lluva et al. [9] presented a similar approach for monitoring older adults in multi-resident environments, either in residential or nursing homes. The system could calculate room-level location using BLE (Bluetooth Low Energy) technology and/or precise positioning using UWB (Ultra Wide Band) in rooms where specific position-related behaviors needed to be detected.

## 3. Methods

The proposed method for extracting routines has the following three main steps:Obtain the location of the user each minute of the day;Derive a feature vector for each of the users, providing information about the activities performed on each weekday;Estimate the routines of the user.

### 3.1. Problem Formulation

In this paper, a routine is defined as the probability of doing a particular activity $a_{i}\in A$, with $i=[1,\ldots ,n_{a}]$, $n_{a}$ being the number of different activities, and *A* being the set of all possible activities, given the weekday *w* (w∈[Monday,…,Sunday]). For instance, one routine can be P(a=Gotogym|w=Monday)=0.8, i.e., on eighty percent of the Mondays, the user went to the gym.

Let us assume that data have been collected for $n_{u}$ users, and for each user $u_{k}$ ($u_{k}\in \Sigma$, $k=[1,\ldots ,n_{u}]$ and Σ being the set of all users), there are data for $n_{d}$ days. We also define each day belonging to the study as $d_{j}$ ($d_{j}\in \Omega ,j=\left[1,\ldots ,n_{d}\right]$ and the set of all days belonging to the study as Ω).

### 3.2. Indoor Localization

For the indoor localization step, it is assumed that there is a radio map D={Λ,L}, where $\Lambda =[\lambda_{1}^{tr},\ldots ,\lambda_{q}^{tr},\ldots ,\lambda_{n_{tr}}^{tr}]$ is a set of training fingerprints and $L=[\tau_{1}^{tr},\ldots ,\tau_{q}^{tr},\ldots ,\tau_{n_{tr}}^{tr}]$ is the set of their associated locations. Each training fingerprint $\lambda_{q}^{tr}$ is composed of $n_{b}$ RSSI values $\lambda_{q}^{tr}=\left[\rho_{1}^{tr}\left(q\right),\ldots ,\rho_{l}^{tr}\left(q\right),\ldots ,\rho_{n_{b}}^{tr}\left(q\right)\right]$, with $n_{b}$ being the number of beacons deployed in the scenario. Therefore, $\rho_{l}^{tr}\left(q\right)$ is the RSSI value obtained for the *q*-th fingerprint and the *l*-th beacon.

In this paper, we work at room level, where each element $\tau_{q}^{tr}$ of the set L is a room ID. We also assume that the user is wearing a device that is able to obtain RSSI measurements from the beacons deployed in the scenario, and that enables the creation of a testing fingerprint $\lambda^{ts}\left(t\right)$ at a particular time moment *t*. A fingerprint $\lambda_{m}^{ts}$ can be obtained each minute, for instance, by estimating the mean of all fingerprints obtained for a particular minute *m*.

Therefore, the indoor localization problem consists of the estimation of the room ID (${\hat{\tau }}_{m}^{ts}$) related to the test fingerprint $\lambda_{m}^{ts}$ obtained at minute *m*, given the radio map D. Figure 1 shows this process. For this purpose, any supervised classifier can be used. For instance, if a 1-NN classifier is used [10], then the localization of the test fingerprint is the most similar using the Euclidean distance in the feature (fingerprint) space of the ones included in the radio map.

Following the proposed procedure, for each day $d_{j}$, it is possible to obtain the location of a user $u_{k}$ every minute, using $L_{j}^{u_{k}}$. Since the location of the user is estimated each minute in a day, the set $L_{j}^{u_{k}}$ has 60∗24=1440 elements.

### 3.3. Feature Vector Extraction

This paper aims to investigate the daily routines of older adults residing in a nursing home. In this scenario, routines are quite stable for a long time, and some activities are performed at approximately the same time for all users. For instance, all the users go to breakfast, have lunch, and have dinner at very similar times when the dining room is open. In addition, all of them stay in their rooms after having lunch and until breakfast the next day. Therefore, detecting this kind of activity is not relevant, all users exhibit similar behavior. Taking this into account, we focus on the following $n_{a}=6$ activities:$A_{1}$ (*Room_M*): To be in the room in the morning.$A_{2}$ (*Physically_M*): To do an activity that needs some level of physical activity in the morning.$A_{3}$ (*Passive_M*): To do an activity that does not need physical activity in the morning.$A_{4}$ (*Room_A*): To be in the room in the afternoon.$A_{5}$ (*Physically_A*): To do an activity that needs some level of physical activity in the afternoon.$A_{6}$ (*Passive_A*): To do an activity that does not need physical activity in the afternoon.

We define Morning as the period after breakfast and before lunch. We also define Afternoon as the period after lunch and before dinner.

For instance, on the one hand, a *Physically_M* (*Physically_A*) activity can include exercising in the gym. On the other hand, a *Passive_M* (*Passive_A*) can be to watch the TV in the common TV room, to sit outside for a while on the terrace, garden, or similar.

From the set of all locations estimated every minute for a user $u_{k}$ and on a given day $d_{j}$, $L_{j}^{u_{k}}$, the feature vector $f_{j}^{u_{k}}$ can be obtained. Figure 2 shows this process. This vector has two elements. The first one is the weekday, and the second one is a vector of $n_{a}$ binary numbers, indicating if the user stayed in the location (or locations) where the corresponding activity was realized. For instance, if a user stayed in their room in the morning (*Room_M*), went to the gym after having lunch (*Physically_A*) and then walked outside (*Passive_A*), and it is a Monday, the feature vector $f_{j}^{u_{k}}$ obtained for the user $u_{k}$ and for the *j*-th day of the study will be expressed as follows:(1)$$f_{j}^{u_{k}}=\left\{Monday,\left[1,0,0,0,1,1\right]\right\}$$

The set $F^{u_{k}}$ is defined as the set with all the feature vectors obtained for the user $u_{k}$ for all days $d_{j}\in \Omega$ of the study.

To determine if a user performed an activity, the following rules are used. Suppose the user stayed more than a defined number of minutes ($n_{m}$) in some of the locations corresponding to a specific activity. In that case, it is assumed that the user performed such an activity. A sliding window of $n_{w}$ minutes is used, with a step of $n_{s}$ minutes.

Figure 3 illustrates this process with an example. It shows a segment of the vector containing the room IDs ($L_{j}^{u_{k}}$) at a specific moment in the morning. In this representation, a value of “1” corresponds to the user’s room (associated with activity $A_{1}$), “2” represents the gym (linked to activity $A_{2}$), and “0” indicates that the user’s location could not be determined. A sliding window of size $n_{w}=10$ with a step size of $n_{s}=5$ is applied to determine whether the user stayed in a specific room. Assuming a threshold of $n_{m}=5$, the first row of the figure shows a case where the user remained in Room 1. The window contains nine occurrences of the value “1”, which exceeds $n_{m}$, leading to the conclusion that $A_{1}=1$. For activity $A_{2}$, which corresponds to staying in the gym (Room 2), the second row of the figure presents a scenario where the number of “2”s within the sliding window is four, which is below $n_{m}$. Therefore, we cannot confirm that the user performed activity $A_{2}$ in this instance. However, the third row demonstrates a situation where the number of “2”s within the window meets or exceeds $n_{m}$, allowing us to conclude that $A_{2}=1$.

### 3.4. Routine Estimation

Given the sets $F^{u_{k}}$, the final step is to estimate the probability of performing a particular activity $a_{i}$ on a given weekday *w*, i.e., $P(a=a_{i}|w=w)$. This can be estimated as follows:(2)$$P\left(a=a_{i}|w=w\right)=\frac{\sum_{j}\alpha \left(f_{j}^{u_{k}}\left(p\right),w\right)}{\sum_{j}\beta \left(d_{j},w\right)}$$
where $f_{j}^{u_{k}}\left(p\right)$ refers to the binary number at the *i*-th position of the vector $f_{j}^{u_{k}}$, and $\alpha (f_{j}^{u_{k}}\left(i\right),w)$ is a function that returns 1 if the *p*-th position of the vector $f_{j}^{u_{k}}$ is equal to 1 and the day $d_{j}$ is *w*, and 0 otherwise. The top portion of Equation (Equation 2) accounts for the number of times that the weekday is $w_{l}$ and the user performed the *i*-th activity. $\beta (d_{j},w)$ is a function returning 1 if the weekday of $d_{j}$ is *w*, and 0 otherwise. The bottom part of Equation (Equation 2) accounts for the number of days in the study being *w*.

Using the aforementioned equation, a matrix $M^{u_{k}}$ is determined for each user by estimating the probabilities of all activities in *A*, given all the weekdays. As only working days are considered in this study, and there are $n_{a}=6$ activities, matrix $M^{u_{k}}$ has 5 rows and 6 columns.

## 4. Dataset

In this paper, the latest version (Version 0.4.0) of the dataset previously published in [1] was used; that paper [1] only explained the data from campaigns 1 to 4, but the same protocol was used to obtain campaigns 6 and 7. In particular, the sixth and seventh data campaigns were used in this paper, comprising almost 5 months of data ($n_{d}=88$). Unfortunately, the caregivers collaborating with us never worked on weekends. Therefore, there were only data from Monday to Friday. These two campaigns were selected to test our proposal, as they involved the same volunteers and represented the longest possible period available with the consistent participants in this dataset. The data can be accessed by [11]. Table 1 presents information on the $n_{u}=6$ volunteers (3 males and 3 females) who participated in this study. In all cases, they were 84 years old or older. They lived in a nursing home, where volunteers F176 and 402E shared a room, while the others each had individual rooms.

The dataset comprised RSSI signals from the BLE beacons installed in the nursing home, measured by a smartwatch worn by the 6 volunteers during their daily routines. A total of 15 BLE beacons were deployed, 1 in each volunteer’s room, 2 in the TV Room, 2 in the dining room, 2 in the gym, 2 on the terrace, and 1 in the therapy room. Figure 4 shows the location of the beacons in the nursing home. The nursing home has 4 floors. The dining room, therapy rooms, and gym are located on the ground floor (0). The TV room and the terrace are located on the first floor. The volunteers’ rooms are distributed on the first, second, and third floors.

The beacons used were model 105 from the company iBKS [12]. Its small size allows it to be deployed in various locations in the environment for monitoring purposes. They can be used with the iBeacon and Eddystone protocols. The manufacturer provides an app to define the configurable parameters. Specifically, they were configured with a transmission period of 200 ms and a transmission power of −4 dBm, all using the iBeacon protocol. Using this configuration, the BLE signal covered all areas within each room, and the battery life was approximately 4 months. The smartwatches worn by the volunteers were the model Smartwatch 3 from Sony, based on the Android Wear 6.0.1 operating system. Their size makes them unobtrusive in the daily activities of the users. The smartwatch firmware was modified to run an app that continuously scanned at the maximum sampling rate allowed by the operating system. They stored the information on a microSD card. The battery life was approximately 10 to 12 h.

No specific directives were given to the volunteers regarding their behavior during the data collection period. They carried on with their daily routines. The same data collection process was followed every day for each volunteer. Every morning, the caregiver placed the smartwatch on the volunteer’s wrist when he/she woke up. The volunteers wore the smartwatch until they returned to their room after dinner, where the caregiver would remove it to charge the battery in the nursing room. Unfortunately, in some cases, the battery may have been discharged before the caregiver was able to remove the smartwatch from the volunteer. On days when the caregiver was not working, no data were obtained.

Activities performed in the gym and the therapy rooms were considered, in this paper, as active ($A_{2}$ and $A_{5}$, see Section 3), as the volunteers participate actively in the performed activities. In contrast, staying in the TV room or going to the terrace were considered passive activities ($A_{3}$ and $A_{6}$, see Section 3), as the volunteers did not perform any physical activity.

The caregivers provided us with an explanation in simple terms about the routine performed by each volunteer. For instance, the following information was provided for user 52EA:He wakes up between 8:00 and 9:30.He has breakfast in the dining room around 9:30. When he finishes breakfast, he goes to his room.He stays in his room (watching TV) until 12:45.He has lunch in the dining room from 12:45 to 13:30. When he finishes lunch, he goes to his room.He stays in his room until 16:20.He goes down to the dining room to have a snack. When he finishes his snack, he goes up to his room.He stays in the room until 19:45.He has dinner in the dining room from 19:45 to 20:30.When he finishes dinner, he goes up to his room.

In some cases, the information provided was not as clear as the one shown for the user 52EA, providing information such as, “Some days, he goes to the gym after having lunch” or “Three days per week, she goes to the therapy room”. Unfortunately, the caregivers did not provide information about the particular days these volunteers performed such activities.

## 5. Experiments and Results

### 5.1. Experimental Set Up

According to how the feature vector $F^{u_{k}}$ was defined in Section 3.3 and considering where the beacons were deployed in the dataset used, *$A_{2}$: Physically_M* and *$A_{5}$: Physically_A* activities were defined as the one performed in the gym and the therapy room. On the other hand, *$A_{3}$: Passive_M* and *$A_{6}$: Passive_M* activities were the ones carried out on the terrace and in the TV room.

The dataset included a radio map produced by the dataset creators by staying several minutes at each location to capture RSSI data. We tested the quality of the radio map by randomly dividing the samples included in the radio map into 70% for training and 30% for validation, obtaining an accuracy of 100% when using a 1NN supervised classifier. Figure 5 shows the confusion matrix obtained in this particular configuration, where 76, 51, 136, 89, 71, and 37 test fingerprints were obtained in the dining room, gym, users’ rooms, TV room, terrace, and therapy room, respectively.

To extract the feature vectors $f_{j}^{u_{k}}$, we tested several values for the parameters $n_{w}$, $n_{s}$, and $n_{m}$. The best results were obtained for the values $n_{w}=60$, $n_{s}=5$, and $n_{m}=10$, but there were no significant differences when using other values such as $n_{w}=30$, $n_{s}=10$, or $n_{m}=15$. This means that the activity $a_{i}$ is considered complete only if a window of 60 min can be found during which the user stayed at least 10 min in the location of the activity.

To determine whether the routines generated for each user accurately reflect their actual behavior, the ground truth information provided to caregivers in text format was transformed into probabilities through human interpretation. These resulting matrices denoted as $G^{u_{k}}$ are shown in the left part of Table 2 and have the same dimensions as the matrices $M^{u_{k}}$ to be estimated. To compare the corresponding pairs of probability matrices, the Mean Squared Error (MSE) metric was utilized.

### 5.2. Localization Results

Figure 6 shows the localization results for the six volunteers. On the right of each subfigure, the weekdays are indicated. The *x* axis represents the hour of the day. Each row of the subfigures represents a day in the life of the volunteer where there are data available. Each of the following colors represents a different location where the user stayed:White: Not possible to estimate the location;Blue: Volunteer’s room;Red: Dining room;Yellow: A place where the volunteer performed a passive activity;Green: A place where the volunteer performed an active activity.

Due to battery limitations, no data were obtained from the late afternoon until the next day. Volunteers always stayed in their rooms during those periods. The results obtained showed that all users returned to the dining room at approximately the same time to have breakfast, lunch, an afternoon snack, and dinner.

According to the results obtained with the proposed approach (see Figure 6a), user 9FE9 tends to perform active activities after having breakfast, although they may perform passive activities on some days. Almost every day, they perform passive activities after having lunch. Some days, they stay in their room for a while. After having dinner, the user tends to participate in active activities. User 9FE9 represents a very active volunteer. In contrast, user 52EA (see Figure 6c) almost always stays in their room. Notably, user 402E (see Figure 6d) had a strong change in routine in the middle of the study period. They used to perform active activities, but there was a point at which the routine shifted to passive ones. This could be due to some health problem of the volunteer or that they decided to pass the time after having lunch on the terrace since, during this period, the weather was good enough to permit this activity, in contrast to the colder days before.

These results demonstrate the effectiveness of our methods in accurately detecting the location of users in a real-world scenario with actual participants. Despite the inherent challenges, such as limited data collection due to battery constraints, the approach successfully identifies key patterns of movement and location transitions throughout the day. This includes detecting periods of activity in shared spaces, such as the dining room, and extended stays in personal rooms. Such precision in localization is very important for applications in real scenarios, where understanding user behavior and routine is essential, particularly in contexts like elderly care, health monitoring, or personalized assistance systems. The ability to work effectively with real people under realistic conditions highlights the practical viability and robustness of the proposed approach.

### 5.3. Routines Extraction Results

The left part of Table 2 shows how the text files providing explanations about the volunteers’ behavior (as observed by the caregivers) have been transformed into routines, i.e., the matrices $G^{u_{k}}$. The right part of Table 2 shows the results obtained using the proposed approach, i.e., the matrices $M^{u_{k}}$. The automatically extracted $M^{u_{k}}$ is expected to be similar to the ground truth information $G^{u_{k}}$.

The routine estimated for user 02A8 (see Table 3) using the proposed methodology reveals a highly stable pattern in the mornings, consistently showing a strong tendency to stay in the room across all weekdays. In the afternoon, the behavior is also mostly sedentary, with high probabilities of staying in the room, but with slightly more variation. On some days, particularly Monday, Wednesday, and Friday, there is a moderate likelihood of engaging in physical activity, while on Tuesday and Thursday, there is a small chance of performing non-physical (calm) activities. Overall, the routine suggests a low-activity profile, especially in the morning, with limited but slightly more diverse behavior in the afternoon.

The estimated routine for user 9FE9 (see Table 4) shows a high consistency across all weekdays, with high probabilities for $\hat{A_{2}}$. Additionally, there is some variation in the morning routine, with occasional involvement in non-physical activities. In the afternoon, the user regularly participates in physical and non-physical activities, especially later in the week (Wednesday to Friday), with high probabilities for both $\hat{A_{5}}$ and $\hat{A_{6}}$. The afternoon routine appears more balanced and active compared to the morning, suggesting a user who maintains a relatively high level of activity throughout the week, especially as the week progresses.

The estimated routine for user 52EA (see Table 5) shows a highly consistent and stable behavior pattern throughout the week. In the morning, the user almost always stays in the room, with very high probabilities for $\hat{A_{1}}$ and negligible likelihood of doing any other activities. In the afternoon, the user also tends to stay in the room, with $\hat{A_{4}}$ consistently having the highest probability across all weekdays. The probabilities for engaging in either physical or calm activities in the afternoon are extremely low or zero, indicating a very sedentary routine both in the morning and afternoon.

User 402E (see Table 6) displays a clear preference for physical activities in the morning, with very high probabilities for $\hat{A_{2}}$ across all weekdays. There is also occasional participation in calm morning activities (e.g., Tuesday and Thursday) but never a preference for staying in the room in the morning. In the afternoon, the routine is more varied but consistently includes physical activity ($\hat{A_{5}}$), often combined with calm activities ($\hat{A_{6}}$). The routine suggests a user with a physically active lifestyle, especially during the early part of the day, and who maintains a high level of engagement in both types of activities in the afternoon.

User 682A (see Table 7) shows a very consistent routine in the morning, where they almost always stay in the room ($\hat{A_{1}}$), with probabilities near or equal to one across all weekdays. There is some variability with minor probabilities for physical and calm activities, but they are never dominant. In the afternoon, the user consistently engages in a calm activity ($\hat{A_{4}}$), again with high probabilities throughout the week. The probabilities for physical activity or staying in the room during the afternoon are very low or zero. Overall, the routine suggests a stable pattern with quiet mornings and structured, calm engagement in the afternoons.

Finally, user F176 (see Table 8) has a routine that is both active and diverse. In the morning, the user predominantly engages in physical activities ($\hat{A_{2}}$), with high probabilities from Monday to Friday. On some days, especially Tuesday and Wednesday, there is also moderate participation in calm activities ($\hat{A_{3}}$), indicating a varied morning pattern. In the afternoon, the user is highly active, with consistent engagement in calm activities ($\hat{A_{4}}$) and physical activities ($\hat{A_{5}}$). Occasionally, other types of activities are also observed (e.g., $\hat{A_{6}}$ on Monday). Overall, F176 follows a rich and balanced routine involving both physical and calm engagement throughout the week.

Figure 7 shows, using bar diagrams, the difference for each user between the estimated probability of a given activity on a weekday and the behavioral information provided by the caregivers. The taller the bar, the higher the discrepancy between the estimated value and the one provided by caregivers. For some users, for instance, user 52EA (see Figure 7c), the probabilities obtained by the proposed approach are very close to the ones derived from the information provided by the caregivers. For some of the other users, for instance, user 682A (see Figure 7e), there is a strong difference in estimating the activity $A_{6}$. This discrepancy can be the result of the following two possible factors: (1) The proposed system was not able to detect this activity since the user performed it at the end of the day when the smartwatch battery was completely drained. (2) The information provided by the caregiver was not entirely accurate, since he/she did not remember the user’s routine well. In this particular case, the localization results shown in Figure 6e suggest that the problem is the first one. Upon analyzing the localization results of user 402E (see Figure 6d), an abrupt change in routine is found, as the user changed from performing active activities after lunch to engaging in passive ones. This change in routine is not reflected in the data provided by the caregivers (see Table 2), which indicate that user 402E always performs active activities after lunch ($A_{5}$).

To compare the results obtained for each user with the probabilities obtained from the behavioral descriptions provided by the caregivers, the MSE between the two matrices is estimated and presented in Table 9. Figure 8 shows a boxplot with the values obtained. In general, the proposed method produces a set of probabilities for performing an activity on a given weekday that aligns with the information provided by the caregivers.

### 5.4. Limitations

The proposed method has several limitations, which are described in the following sections.

#### 5.4.1. Days Without Data

One of the most important limitations of this study is the number of days without data. Unfortunately, there were only data when the caregivers were working, as they were essential to implementing the acquisition protocol. In addition, on some days, the volunteers were not able to perform the required protocol due to different physical or mental health issues. Another possible problem is that when the caregiver is away on a non-working day, resulting in no data being collected for this day and the following one, as no one switched on the smartphone at night to charge the device’s battery. Table 10 shows the number of days and percentage of the total number of days on which data were recorded ($n_{d}=88$).

#### 5.4.2. Limitation of Android Wear Operating System

Another problem regarding missing data is that the Android Wear operating system did not allow us to obtain as many RSSI measurements as desired. For instance, if no motion was detected by the smartwatch, no data were obtained. This produced some periods without the possibility of estimating the position of the user.

#### 5.4.3. Smartwatch Battery Life

The smartwatch battery lasted 10–12 h per day. Therefore, on several occasions, the battery life was not enough to obtain data between the afternoon snack and dinner. For this reason, most of the discrepancies detected (see Figure 7) were concentrated in the afternoon period.

#### 5.4.4. Ground Truth Information Provided by Caregivers

Despite the excellent efforts of the caregivers who collaborated in this study—without which this study would not have been possible—they are still human, and they may not remember exactly what the users did every day of the study. Therefore, although we used the information provided by them as the ground truth, in some cases, this might not be entirely accurate. Unfortunately, there is no way to verify this.

## 6. Conclusions and Future Work

This paper has introduced a methodology for extracting and analyzing the daily routine of a group of volunteers residing in a nursing home from raw Received Signal Strength Indicator (RSSI) data. The volunteers, who were older adults, wore a smartwatch that captured RSSI signals released from a set of Bluetooth Low Energy (BLE) beacons strategically deployed throughout the nursing home. The proposed methodology unfolds through the following three key steps: (a) minute-by-minute localization estimation for each user using an indoor localization fingerprinting-based method, (b) creation of a feature vector elucidating the activities undertaken by each volunteer at various intervals throughout the day, and (c) estimation of the probability of a specific activity being performed by each volunteer based on the day of the week. The calculated probabilities were juxtaposed with those deduced from the caregivers’ insights into the volunteers’ routines. Despite the inherent limitations of this study, the results obtained provide the most pertinent routines of the volunteers.

Subsequent research efforts should concentrate on the following two primary avenues: (1) employing a device with an extended battery life to ensure comprehensive day-long data acquisition, thereby mitigating the caregivers’ involvement in the data acquisition process, and (2) obtaining more detailed information from caregivers to establish a more precise ground truth.

## Figures and Tables

**Figure 1 sensors-25-02745-f001:**
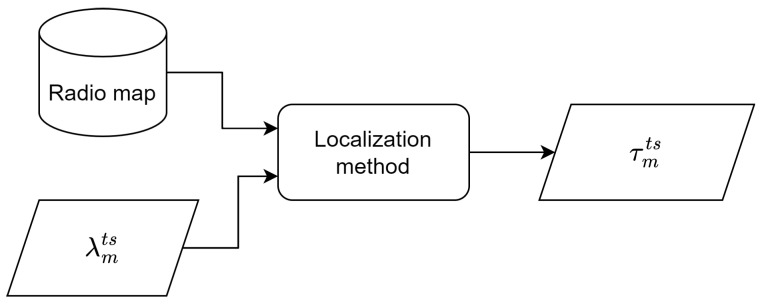
Method for obtaining the volunteer’s location every minute $\tau_{m}^{ts}$, given the fingerprint obtained for this minute $\lambda_{m}^{ts}$.

**Figure 2 sensors-25-02745-f002:**
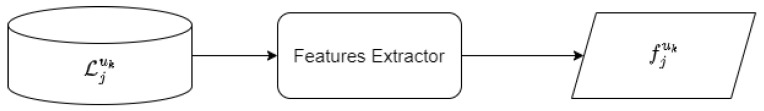
From the set of locations obtained every minute for each user, a feature vector is obtained.

**Figure 3 sensors-25-02745-f003:**
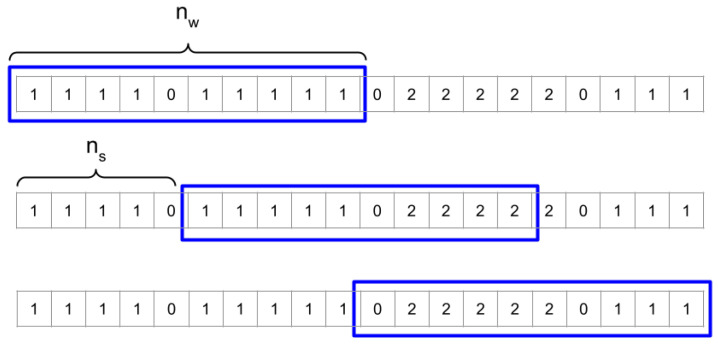
An illustrative example of how to obtain feature vectors $f_{j}^{u_{k}}$ from location ones $L_{j}^{u_{k}}$. In this example, $n_{w}=10$ (blue boxes) and $n_{s}=5$. See text for explanation.

**Figure 4 sensors-25-02745-f004:**
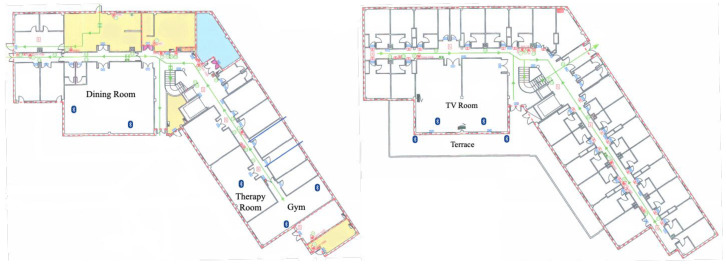
Locations of beacons deployed in nursing home. Left: ground floor (0), right: first floor.

**Figure 5 sensors-25-02745-f005:**
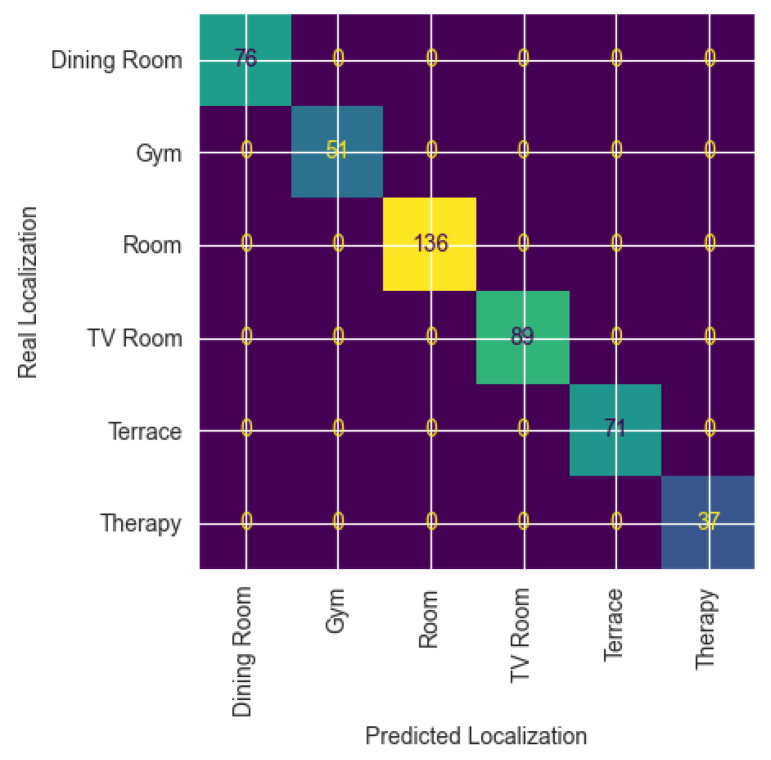
Confusion matrix obtained by evaluating accuracy of test samples used to assess performance of the localization system.

**Figure 6 sensors-25-02745-f006:**
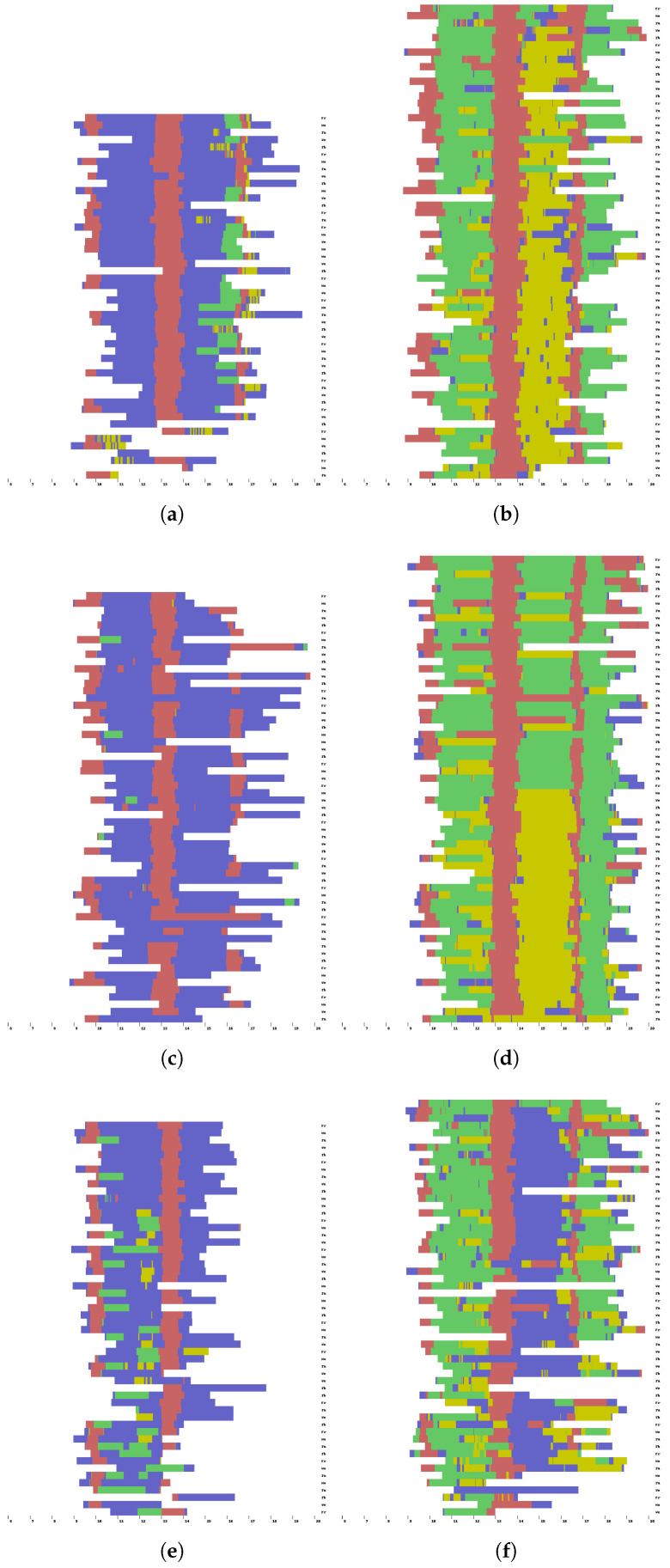
Localization obtained for six volunteers. (**a**) User 02A8; (**b**). User 9FE9; (**c**) User 52EA; (**d**) User 402E; (**e**) User 682A; (**f**) User F176.

**Figure 7 sensors-25-02745-f007:**
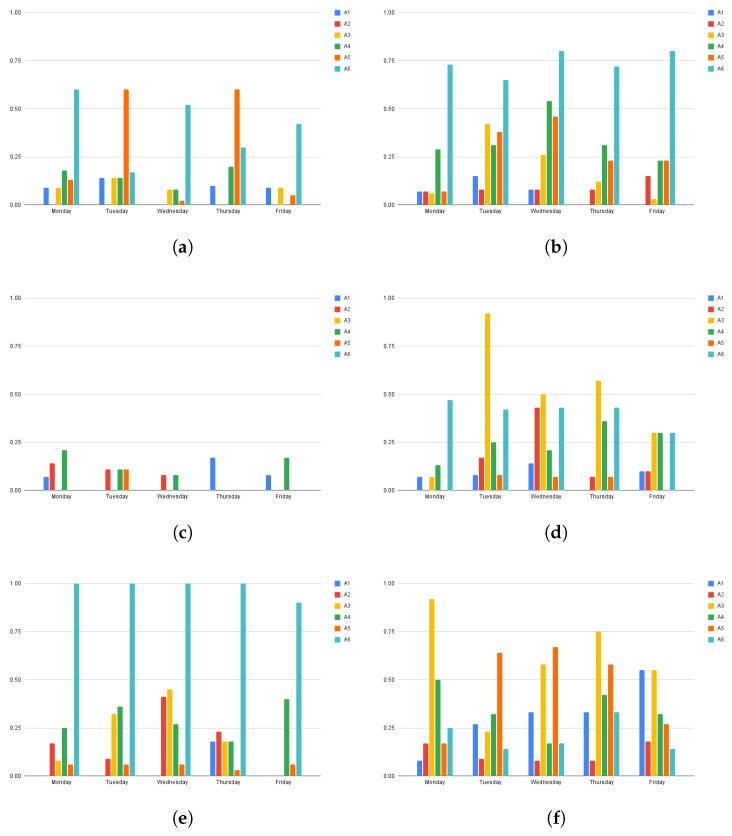
Differences between estimated probability of a given activity on a weekday and behavioral information provided by caregivers of different users. (**a**) User 02A8; (**b**) User 9FE9; (**c**) User 52EA; (**d**) User 402E; (**e**) User 682A; (**f**) User F176.

**Figure 8 sensors-25-02745-f008:**
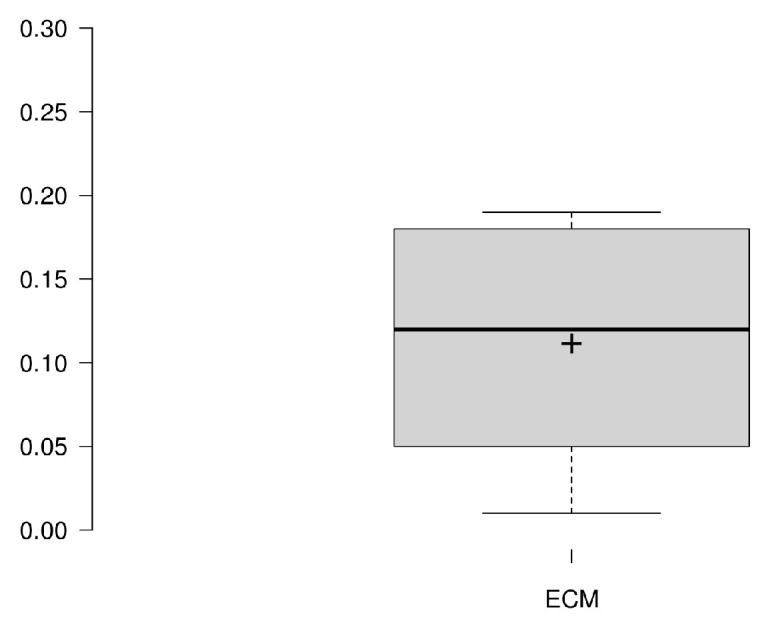
Boxplot of MSE between matrices $G^{u_{k}}$ and $M^{u_{k}}$.

**Table 1 sensors-25-02745-t001:** Volunteers’ information. 1: Walks with a walking frame. 2: Walks without technical assistance. 3: Walks with a cane. 4: Slight dependency. 5: Does not climb stairs. 6: Uses lift. 7: Walks up and down stairs.

ID	Age	Gender	Height	Weight	Observations
52EA	85	M	174	83	1, 4, 5, 6
F176	85	F	165	64	2, 4, 6
402E	84	F	172	75	2, 4, 6
9FE9	86	F	159	72	1, 4, 6
682A	84	M	160	63	2, 4, 6
02A8	89	M	156	65	3, 4, 7

**Table 2 sensors-25-02745-t002:** Left, probabilities of performing the different activities ($A_{i}$) according to the information provided by caregivers ($G^{u_{k}}$). Right, estimated probabilities ($\hat{A_{i}}$) obtained using the proposed methodology ($M^{u_{k}}$).

Volunteer	Weekday	$A_{1}$	$A_{2}$	$A_{3}$	$A_{4}$	$A_{5}$	$A_{6}$	$\hat{A_{1}}$	$\hat{A_{2}}$	$\hat{A_{3}}$	$\hat{A_{4}}$	$\hat{A_{5}}$	$\hat{A_{6}}$
02A8	Monday	1	0	0	1	0.6	0.6	0.91	0	0.09	0.82	0.73	0
	Tuesday	1	0	0	1	0.6	0.6	0.86	0	0.14	0.86	0	0.43
	Wednesday	1	0	0	1	0.6	0.6	1	0	0.08	0.92	0.58	0.08
	Thursday	1	0	0	1	0.6	0.6	0.9	0	0	0.8	0	0.3
	Friday	1	0	0	1	0.6	0.6	0.91	0	0.09	1	0.55	0.18
9FE9	Monday	0	1	0.2	0	1	0.2	0.07	0.93	0.14	0.29	0.93	0.93
	Tuesday	0	1	0.2	0	1	0.2	0.15	0.92	0.62	0.31	0.62	0.85
	Wednesday	0	1	0.2	0	1	0.2	0.08	0.92	0.46	0.54	0.54	1
	Thursday	0	1	0.2	0	1	0.2	0	0.92	0.08	0.31	0.77	0.92
	Friday	0	1	0.2	0	1	0.2	0	0.85	0.23	0.23	0.77	1
52EA	Monday	1	0	0	1	0	0	0.93	0.14	0	0.79	0	0
	Tuesday	1	0	0	1	0	0	1	0.11	0	0.89	0.11	0
	Wednesday	1	0	0	1	0	0	1	0.08	0	0.92	0	0
	Thursday	1	0	0	1	0	0	0.83	0	0	1	0	0
	Friday	1	0	0	1	0	0	0.92	0	0	0.83	0	0
402E	Monday	0	1	0	0	1	1	0.07	1	0.07	0.13	1	0.53
	Tuesday	0	1	0	0	1	1	0.08	0.83	0.92	0.25	0.92	0.58
	Wednesday	0	1	0	0	1	1	0.14	0.57	0.5	0.21	0.93	0.57
	Thursday	0	1	0	0	1	1	0	0.93	0.57	0.36	0.93	0.57
	Friday	0	1	0	0	1	1	0.1	0.9	0.3	0.3	1	0.7
682A	Monday	1	0.5	0	1	0.06	1	1	0.33	0.08	0.75	0	0
	Tuesday	1	1	0.5	1	0.06	1	1	0.91	0.18	0.64	0	0
	Wednesday	1	0.5	0	1	0.06	1	1	0.09	0.45	0.73	0	0
	Thursday	1	0.5	0	1	0.06	1	0.82	0.27	0.18	0.82	0.09	0
	Friday	1	0.5	0	1	0.06	1	1	0.5	0	0.6	0	0.1
F176	Monday	0	1	1	0.5	1	0.5	0.08	0.83	0.08	1	0.83	0.25
	Tuesday	0	1	0.5	0.5	1	0.5	0.27	0.91	0.73	0.82	0.36	0.64
	Wednesday	0	1	1	0.5	1	0.5	0.33	0.92	0.42	0.67	0.33	0.33
	Thursday	0	1	1	0.5	1	0.5	0.33	0.92	0.25	0.92	0.42	0.83
	Friday	0	1	1	0.5	1	0.5	0.55	0.82	0.45	0.82	0.73	0.64

**Table 3 sensors-25-02745-t003:** Estimated routine for user 02A8 based on dominant activities.

Weekday	Morning Activity	Afternoon Activity
Monday	Stay in the room	Stay in the room, possible physical activity
Tuesday	Stay in the room	Stay in the room, possible calm activity
Wednesday	Stay in the room	Stay in the room, possible physical activity
Thursday	Stay in the room	Stay in the room, possible calm activity
Friday	Stay in the room	Stay in the room, possible physical activity

**Table 4 sensors-25-02745-t004:** Estimated routine for user 9FE9 based on dominant activities.

Weekday	Morning Activity	Afternoon Activity
Monday	Physical activity	Physical and calm activity
Tuesday	Physical activity, some calm	Calm and physical activity
Wednesday	Physical activity, some calm	Calm and physical activity
Thursday	Physical activity	Physical and calm activity
Friday	Physical activity	Calm and physical activity

**Table 5 sensors-25-02745-t005:** Estimated routine for user 52EA based on dominant activities.

Weekday	Morning Activity	Afternoon Activity
Monday	Stay in the room	Stay in the room
Tuesday	Stay in the room	Stay in the room
Wednesday	Stay in the room	Stay in the room
Thursday	Stay in the room	Stay in the room
Friday	Stay in the room	Stay in the room

**Table 6 sensors-25-02745-t006:** Estimated routine for user 402E based on dominant activities.

Weekday	Morning Activity	Afternoon Activity
Monday	Physical activity	Physical activity, some calm activity
Tuesday	Physical and calm activity	Physical and calm activity
Wednesday	Physical activity	Physical and calm activity
Thursday	Physical activity	Physical and calm activity
Friday	Physical activity	Physical and calm activity

**Table 7 sensors-25-02745-t007:** Estimated routine for user 682A based on dominant activities.

Weekday	Morning Activity	Afternoon Activity
Monday	Stay in the room	Stay in the room
Tuesday	Stay in the room, Physical activity	Stay in the room
Wednesday	Stay in the room, some calm activity	Stay in the room
Thursday	Stay in the room	Stay in the room
Friday	Stay in the room	Stay in the room

**Table 8 sensors-25-02745-t008:** Estimated routine for user F176 based on dominant activities. Act., Phy., and SiR stand for “Activity”, “Physical”, and ”Stay in the room”, respectively.

Weekday	Morning Activity	Afternoon Activity
Monday	Phy. act.	SiR, Phy. act., some calm act.
Tuesday	Phy. and calm act.	SiR, calm and some Phy. act.
Wednesday	Phy. and some calm act.	SiR, some calm and some Phy. act.
Thursday	Phy. act.	SiR, calm and some Phy. act.
Friday	Phy. and calm act.	SiR, calm and Phy. act.

**Table 9 sensors-25-02745-t009:** MSE between matrices $G^{u_{k}}$ and $M^{u_{k}}$.

Volunteer	MSE
02A8	0.05
402E	0.12
52EA	0.01
682A	0.18
9FE9	0.12
F176	0.19

**Table 10 sensors-25-02745-t010:** Number of days with data of each user.

Volunteer	Days	%
02A8	51	58%
9FE9	66	66%
52EA	60	60%
402E	65	65%
682A	55	55%
F176	58	58%

## Data Availability

Data are accessible by [1].

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
