# Peer review of "Extracting Daily Routines from Raw RSSI Data"

_sensors, 2025, doi:10.3390/s25092745_

Round 1
Reviewer 1 Report
Comments and Suggestions for Authors
The paper presents a methodology for extracting daily routines of elderly individuals in a care home using RSSI data from BLE beacons and smartwatches. While the work is innovative and addresses a relevant real-world problem, several areas require improvement to enhance clarity, rigor, and reproducibility:
1.In the abstract, you mention that the study estimates the probability of a specific activity, but lack of experiment result.
2.The transition between Sections 3.2 (Indoor Localization) and 3.3 (Feature Extraction) is abrupt. A summary figure outlining the end-to-end pipeline would help readers visualize the workflow.
3.Add a schematic diagram of the nursing home and beacon deployment.
4. Dataset is generated based on six volunteers, I wonder whether it is reliable to derive the routine from a small dataset.
5. There is no data to validate your localization method in figure 3. How can you ensure the ground truth of localization.
Author Response
Please see the Rebuttal.pdf file.

Reviewer 2 Report
Comments and Suggestions for Authors
This paper proposed a method to obtain the daily routines of older adult volunteers. It collected RSSI measurements to estimate the locations and recognize the activities. It is a very interesting work to combine location, activity recognition and temporal analysis.
Figure 3 shows the localization results for the six volunteers. However, the accuracy of location is not given clearly. Are there any location errors leading to incorrectly determine the room or the transition from one room to the other room? The accuracy of RSSI based location highly depends on the environmental stability.
The study estimates the probability of a user undertaking a specific activity on a given weekday. The qualitative analysis should provide more details and insights.
Author Response
Please see the Rebuttal.pdf file.
